# Drug-Induced Sarcoid-like Reactions Associated to Targeted Therapies and Biologic Agents

**DOI:** 10.3390/diagnostics15131658

**Published:** 2025-06-29

**Authors:** Federica Andolfi, Luca Caffarri, Matilde Neviani, Silvia Rubini, Dario Andrisani, Filippo Gozzi, Bianca Beghé, Enrico Clini, Roberto Tonelli, Stefania Cerri

**Affiliations:** 1Department of Medical and Surgical Sciences, University of Modena and Reggio Emilia, 41121 Modena, Italy; lucacaffarri@gmail.com (L.C.); matilde.neviani@gmail.com (M.N.); silvia.rubini87@gmail.com (S.R.); bianca.beghe@unimore.it (B.B.); enrico.clini@unimore.it (E.C.); roberto.tonelli@unimore.it (R.T.); stefania.cerri@unimore.it (S.C.); 2Respiratory Disease Unit, University Hospital of Modena—Policlinico, 41124 Modena, Italy; andrisanidario@gmail.com (D.A.); fillo.gzz@gmail.com (F.G.); 3Center for Rare Lung Disease, University Hospital of Modena—Policlinico, 41124 Modena, Italy

**Keywords:** sarcoidosis, drug-induced sarcoid like reactions, immune checkpoint inhibitors, TNF-α antagonists, BRAF inhibitors, monoclonal antibodies

## Abstract

**Background**: Sarcoidosis is a multisystem inflammatory disease characterized by the immune-mediated formation of non-necrotizing epithelioid granulomas. Several commonly used medications can induce similar granulomatous reactions, known as drug-induced sarcoid-like reactions (DISRs), which closely mimic sarcoidosis. Despite their specificity in targeting molecular pathways, certain therapies—particularly targeted treatments—have increasingly been linked to DISRs. **Methods**: This narrative review was based on a PubMed search using the terms “SARCOID LIKE REACTION” and “DRUG”. A cross-check was performed with “SARCOID” combined with each identified drug to identify misclassified cases. Drugs with limited evidence or weak pathogenetic plausibility were excluded, leaving only molecularly targeted therapies for consideration. Sources included case reports, case series, and reviews selected based on their clinical and scientific relevance, without any restrictions on time or language. **Results**: In light of the available data, five main pharmacological groups were found to be associated to DISR: immune checkpoint inhibitors, TNF-α antagonists, BRAF inhibitors, monoclonal antibodies, and miscellaneous agents. Each group has distinct mechanisms of action and clinical indications, which likely affect the frequency, presentation, and timing of DISRs. **Conclusions**: Diagnosing DISRs is challenging, and a structured approach is crucial for differentiating them from other conditions. To support clinicians, we propose a diagnostic algorithm to guide decision-making in suspected cases. Management should be individualized, as most DISRs either resolve spontaneously or improve after the discontinuation of the causative drug. Important factors influencing therapeutic decisions include the severity of the underlying disease, the availability of alternative treatments, and the extent of DISR manifestations.

## 1. Introduction

Sarcoidosis is a multisystem inflammatory disease with an immune-mediated pathogenesis, characterized by a broad spectrum of clinical manifestations [1].

The key feature of the disease is the formation of epithelioid non-necrotizing granulomas (Figure 1), which result from a complex interplay between the innate and adaptive immune systems in genetically predisposed individuals [2].

Interestingly, several commonly used drugs can induce the formation of similar granulomas, leading to clinical manifestations that closely resemble sarcoidosis. These reactions are known as drug-induced sarcoid-like reactions (DISRs) [3].

Despite the specificity of targeted therapies in modulating precise molecular pathways, these treatments, among others, have been increasingly implicated in the development of DISRs.

DISRs are characterized by the formation of non-caseating granulomas, which, upon histological examination, resemble those observed in classical sarcoidosis [4].

Although the exact immunological mechanisms driving the formation of these structures remain unclear, one widely supported hypothesis suggests that they result from an aberrant immune response in genetically predisposed individuals. In particular, in DISRs associated with targeted therapies, this immune response appears to be triggered by two main mechanisms:Excessive immune response: Certain drugs can induce a significant and uncontrolled increase in immune activity, as observed with checkpoint inhibitors and BRAF inhibitors. These medications can lead to excessive activation of T cells, including Th1 and Th17, ultimately promoting granuloma formation [5,6].Paradoxical immune activation: Some anti-inflammatory medications can alter the immune profile, leading to the unexpected activation of pathways involved in granuloma formation. For example, despite their role in treating inflammatory conditions, both anti-TNF agents and Rituximab have been reported to induce granuloma development in some patients [7,8].

No specific immunological traits have been identified that clearly distinguish sarcoid-like reactions from sarcoidosis at the histological level, and little is known about the personal factors that predispose individuals to the development of the disease. Several studies suggest that circulating Th17 lymphocytes prior to disease onset may play a pivotal role in pathogenesis, although the evidence supporting this remains limited [8]. This histological similarity is mirrored by clinical resemblance, further complicating the diagnostic process. Both conditions are typically diagnoses of exclusion, and clinical suspicion often arises from radiological findings and characteristic signs that closely mimic sarcoidosis. Consequently, the lack of definitive diagnostic markers and overlapping clinical and histopathological features hinder the collection of accurate epidemiological data, making it difficult to estimate true incidence and prevalence. Given the rising use of targeted therapies and the frequent misdiagnosis of these reactions, often resulting in unnecessary or inappropriate management, this review aims to provide a summary of current knowledge on sarcoid-like reactions specifically associated with targeted therapies.

## 2. Materials and Methods

This narrative review was conducted using the PubMed database. The initial search was performed using a primary string composed of the keywords “SARCOID LIKE REACTION” and “DRUG”, which yielded 520 records and provided a base of relevant articles and a list of associated drugs. A cross-check was then conducted by entering the keyword “SARCOID” in combination with the name of each selected drug, in order to retrieve cases that had been mistakenly classified as sarcoidosis. An initial screening was performed to exclude articles involving drug classes for which only a very limited number of cases had been reported, as well as those lacking a plausible pathogenetic link to sarcoid-like reactions. This was performed to avoid including instances where a causal association between the drug and the reaction could not be reasonably supported, or where the presence of concomitant sarcoidosis appeared more likely. A second screening was then conducted, leaving only molecularly targeted therapies. Following duplicate removal, 294 articles were retained for the final review, including single case reports, case series, and review articles. Data were obtained in the form of single case reports, case series, and reviews. No temporal or language filters were applied. Articles were selected based on their clinical and scientific relevance to the topic.

## 3. Targeted Therapies Associated with Drug-Induced Sarcoid-like Reaction

### 3.1. Immune Checkpoint Inhibitors

We identified five agent groups linked to DISR. The first group in the order of importance consisted of immune checkpoint inhibitors (ICIs). ICIs are monoclonal antibodies with antitumor activity. They exert their effect by modulating the immune system through the blockade of inhibitory pathways that downregulate immune system activities. In healthy individuals, these regulatory mechanisms help maintaining immune responses within a physiologic range, protecting the host from autoimmunity. By blocking immune checkpoints, ICIs eliminate the inhibitory signals that regulate T-cell activation, enabling tumor-reactive T lymphocytes to bypass these regulatory mechanisms and initiate an effective antitumor response [9].

ICIs target two main pathways: the cytotoxic T-lymphocyte-associated protein 4 (CTLA-4) and the programmed death-1 (PD-1) (Figure 2). The former is inhibited by Ipilimumab, the first drug in this class, which was approved in 2011 for metastatic melanoma. The latter is targeted by Nivolumab and Pembrolizumab, which bind to the PD-1 receptor expressed by T lymphocytes, while Atezolizumab, Durvalumab, and Avelumab bind to the PD-L1 ligand that is present on cancer cells [10].

While ICIs offer a significant therapeutic advantage by enhancing the immune system’s ability to target and destroy cancer cells, this heightened immune response can also lead to a range of immune-related adverse events (irAEs). The most frequent irAEs for CTLA-4 inhibitors include colitis, hypophysis, and skin rash, while PD-1 inhibitors are more commonly associated with pneumonitis, hypothyroidism, and vitiligo. Typically, PD-1 and PD-L1 inhibitors are better tolerated than CTLA-4 inhibitors [11].

Among these side effects, DISRs are also described (109 cases). Pembrolizumab is the most reported ICI associated with DISRs. Of the 42 cases analyzed for this review, 24 involved Pembrolizumab administered for the treatment of metastatic melanoma and 7 for non-small cell lung cancer. The average time of onset for DISRs was 7.32 months; the earliest case occurred two weeks after drug administration, while the latest appeared 24 months post-treatment. Symptoms were present in just over half of the cases (54%), with the most common being subcutaneous nodules, erythematous papules, dry cough, and fatigue. In 19 cases, no symptoms were reported, and in some reports, symptoms were unknown. Radiological findings were abnormal in nearly all cases, showing lymphadenopathy, primarily hilar and mediastinal, but also affecting other regions such as gastric, retroperitoneal, and cervical lymph nodes, as well as multiple bilateral pulmonary nodules. It is reasonable to hypothesize that, in asymptomatic cases (46%), the detection of radiological abnormalities may have been due to the routine radiological follow-up of patients undergoing ICI therapy for malignancies. This aspect may pose a challenge, as it raises the issue of differential diagnosis between DISRs and disease progression/recurrence.

In terms of treatment, 6 cases were managed with corticosteroids, while in 14 cases, the ICI was simply discontinued. In three cases, steroid therapy was combined with the discontinuation of the drug. In most cases (76% of the cases with known outcome), these therapeutic strategies led to the resolution of the sarcoid-like reactions [12,13,14,15,16,17,18,19,20,21,22,23,24,25,26,27,28,29,30,31,32,33,34,35,36,37,38,39,40,41,42,43,44,45].

Other ICIs associated with sarcoid-like reactions include Nivolumab (21 cases) and Ipilimumab (20 cases), both used alone or in combination (16 cases). In general, these cases present similar radiological findings, diagnostic modalities, treatment, and resolution to those observed with Pembrolizumab. Regarding the timing of onset, Nivolumab and the combination of Nivolumab and Ipilimumab showed a shorter average onset time, specifically 3.96 months (standard deviation: 2.82) and 4.15 months (standard deviation: 3.99), respectively. DISRs related to Nivolumab and Ipilimumab also included the onset of dyspnea in five and six cases, respectively, a symptom not present in reactions associated with Pembrolizumab or the combination of the two drugs [4,19,22,36,43,44,45,46,47,48,49,50,51,52,53,54,55,56,57,58,59,60,61,62,63,64,65,66,67,68,69,70,71,72,73,74,75,76,77,78,79,80,81,82,83,84,85,86,87,88,89].

Additionally, the literature reports a few cases of DISRs caused by Atezolizumab (four cases), Avelumab (one case), Durvalumab (three cases), Sintilimab (one case), and the combination of Dabrafenib/Trametinib (one case) [36,90,91,92,93,94,95,96,97,98] (Table 1).

### 3.2. Tumor Necrosis Factor Alpha Antagonists

The second group identified included tumor necrosis factor alpha (TNF-α) antagonists. TNF-α is a cytokine primarily produced by macrophages that is involved in systemic inflammation and part of a group of mediators that stimulates the acute phase reaction. It can also be produced by CD4+ T lymphocytes, neutrophils, mast cells, eosinophils, and neurons. In physiologic quantities, TNF-α regulates cell proliferation and apoptosis and enhances the activity and recruitment of numerous inflammatory cells.

Anti-TNF-α drugs are biological agents that antagonize this acute phase molecule through various mechanisms. Introduced in 1999, key examples include infliximab, adalimumab, etanercept, golimumab, and certolizumab [99]. These medications are used to treat rheumatologic conditions such as rheumatoid arthritis, ankylosing spondylitis, and psoriatic arthritis, as well as chronic inflammatory diseases like Crohn’s disease, ulcerative colitis, and psoriasis. They selectively bind to TNF-α, forming stable complexes that prevent its interaction with membrane receptors TNFR1 (p55) and TNFR2 (p75), thereby modulating the biological responses it induces or regulates [99,100]. Despite being highly selective agents, these drugs have been associated with various adverse events, including granulomatous infections (such as tuberculosis, leprosy, and leishmaniasis), malignancies (such as lymphoma), hematologic disorders (anemia and pancytopenia), demyelinating conditions, worsening congestive heart failure, and hypersensitivity reactions [101,102]. (Figure 3).

Even if anti-TNF-α biologics are recommended as third-line therapy for sarcoidosis, paradoxically, some patients have experienced worsening of pre-existing sarcoidosis, such as uveitis or cutaneous sarcoidosis [8]. Moreover, numerous reports have described the development of the same reactions in patients without a prior history of sarcoidosis following treatment with TNF-α inhibitors, that resolved after the discontinuation of the therapy [8]. These manifestations can be classified as drug-induced sarcoid-like reactions (DISRs).

In this review, we identified 294 reports including a total of 111 cases of TNF-α induced sarcoid-like reactions, with etanercept being the most frequently represented drug (57 cases, including 2 cases associated with adalimumab and 1 case associated with infliximab), followed by adalimumab (30 cases) and infliximab (21 cases). Two cases associated with certolizumab and golimumab are also described.

The etiology of anti-TNF-α sarcoid-like reaction remains unclear and challenging to explain. It is hypothesized that anti-TNF-α-induced sarcoidosis may result from imbalances involving TNF receptor 2 (TNFR2) and T-regulatory cells. Normally, the activation of TNFR2 (p75) by TNF-α initiates an inhibitory pathway that modulates Treg cells. Consequently, the administration of an anti-TNF-α therapy blocks the TNFR2 pathway, leading to a significant increase in Treg cells, which are crucial regulators of T-cell differentiation. Furthermore, the imbalance between Th1 and Th2 responses contributes to granuloma formation [8].

In the case of Etanercept, the pathogenesis can be linked to its ability to bind lymphotoxin-α (LT-α). It has been demonstrated that this molecule can independently interact with TNF-α receptors, contributing to the formation of infectious granulomas. Combined with the fact that polymorphisms in LT-α have also been linked to an increased incidence of erythema nodosum in sarcoidosis patients, these factors may explain how etanercept-induced alterations in LT-α signaling could play a role in the pathogenesis of DISRs [8].

Among the patients treated with etanercept, rheumatoid arthritis was the most common underlying condition (30 cases), followed by ankylosing spondylitis (10 cases) and psoriatic arthritis and psoriasis (9 cases) [103,104,105,106,107,108,109,110,111,112,113,114,115,116,117,118,119,120,121,122,123,124,125,126,127,128,129,130,131,132,133,134,135]. Other identified conditions included juvenile idiopathic arthritis and Sjögren’s syndrome. Symptoms typically manifested 2.5 years after starting anti-TNF-α therapy. The onset of sarcoid-like symptoms varied widely, ranging generally from 3 weeks to 6 years. One case was also reported where symptoms appeared after 10 years of treatment, and another where they emerged after 17 years of etanercept therapy.

Nearly half of the cases (25 patients) of DISRs associated with etanercept presented with respiratory symptoms, such as dry cough, dyspnea, and fatigue. Twelve patients exhibited cutaneous manifestations, including erythematous nodules or erythroderma, while two cases presented solely with general symptoms such as malaise, low-grade fever, and weight loss. Etanercept, similar to adalimumab and infliximab, has been closely associated with ocular manifestations, including unilateral or bilateral uveitis (12 cases). Additionally, three asymptomatic patients experienced exclusively progressive renal function deterioration.

Radiological changes were observed in 64.9% of patients on chest X-ray or CT scans, including bilateral hilar mediastinal lymphadenopathy, with or without parenchymal involvement. These changes included nodular or reticulonodular infiltrates, ground-glass opacities, and crazy-paving patterns. One case involved pleural effusion with a reticulonodular pattern. Notably, radiological alterations were also present in patients that did not manifest specific respiratory symptoms. In just nine cases, no radiological alterations were found.

Our review revealed that the majority of patients treated with anti-TNF-α who developed DISR exhibited pulmonary involvement, in contrast to other drug categories, such as BRAF inhibitors, where cutaneous involvement was the most common symptom. This observation could be of interest given that, in pulmonary sarcoidosis, alveolar macrophages spontaneously produce excessive amounts of TNF-α, along with interleukins IL-1, IL-2, IL-6, and IL-10, implicating these cells in the formation of non-caseating granulomas. Further investigation is needed to confirm this and eventually demonstrate how it correlates with the local activation of paradoxical mechanisms that drive granulomatous inflammation [8].

The diagnosis of DIRS is typically confirmed by the histopathological identification of non-caseating granulomas, except in cases of ocular uveitis, where diagnosis is based on clinical and fundoscopic evaluation. Transbronchial biopsy or lymph node sampling via EBUS or mediastinoscopy confirmed 27 diagnoses, while skin biopsies provided diagnostic confirmation in 12 cases. Renal, muscle, and salivary gland biopsies have also been reported.

The vast majority of cases were managed by discontinuing the drug (75%, 45 cases), while in 12 cases etanercept was continued. Corticosteroid therapy was the most commonly employed treatment, used in 34 cases, administered either systemically or topically in cases of uveitis. Additionally, the use of other immunosuppressive therapies is also reported, such as methotrexate (six cases), as well as azathioprine, cyclosporine, and leflunomide (one case each).

Almost all cases achieved clinical resolution following treatment, with only one case showing persistent mediastinal lymphadenopathy on radiological follow-up at 3 months.

DIRS associated with other anti-TNFα agents, such as adalimumab [126,128,132,136,137,138,139,140,141,142,143,144,145,146,147,148,149,150,151,152,153] and infliximab [129,154,155,156,157,158,159,160,161,162,163,164,165], exhibited similar clinical and radiological characteristics to those observed with etanercept. Pulmonary and/or mediastinal involvement, along with cutaneous manifestations, were the most reported features. Infliximab was associated with only two cases of uveitis and adalimumab with seven cases of uveitis. The onset of symptoms occurred after an average of 3.6 years for infliximab and 1.3 years for adalimumab.

A case of DIRS secondary to golimumab therapy for ankylosing spondylitis was also reported, presenting with bilateral granulomatous anterior uveitis and a case secondary to certolizumab therapy administered for rheumatoid arthritis, characterized by cutaneous papules. Both cases were resolved following the discontinuation of the biologic therapy and initiation of corticosteroid treatment [126,166] (Table 2).

### 3.3. BRAF Inhibitors

The third group consisted of B-Raf proto-oncogene, serine/threonine kinase (BRAF) inhibitors. BRAF inhibitors are targeted therapies that exert antitumor effects by specifically inhibiting the mutant BRAF protein, which is produced as a result of activating mutations in the BRAF gene. The most prevalent of these mutations is V600E, commonly found in metastatic melanoma, non-small cell lung cancer, and anaplastic thyroid carcinoma. This mutation leads to the continuous activation of the MAPK (mitogen-activated protein kinase) signaling pathway, driving uncontrolled cellular growth and proliferation. By targeting this pathway, BRAF inhibitors block the aberrant signaling that is central to tumor progression [167].

To further inhibit this pathway and prevent reactivation via MEK, a kinase enzyme that phosphorylates and activates ERK (another kinase downstream in the pathway), BRAF inhibitors are often combined with MEK inhibitors such as trametinib and cobimetinib. This combination provides a more comprehensive blockade of the MAPK cascade, enhancing therapeutic efficacy [168].

BRAF inhibitors are associated with various adverse effects, including fever, skin reactions, and liver toxicity [169]. The frequent occurrence of cutaneous manifestations, which can range from photosensitivity to keratoacanthomas and squamous cell carcinoma, highlights the critical role of the MAPK signaling pathway in maintaining skin homeostasis. The inhibition of this pathway, particularly due to the activation of germline mutations in RAS, disrupts normal skin function, resulting in a higher incidence of skin-related side effects [170,171]. In addition to these more common toxicities, BRAF inhibitors have also been linked to immune dysregulation reactions, particularly DISRs. As noted in several recent studies, these reactions can be attributed to the immunomodulatory effects of BRAF and MEK inhibitors (Figure 4). These agents create an immune-stimulatory microenvironment by altering cytokine profiles and enhancing the expression of immune-stimulatory molecules, such as CD40L and IFNγ, while concurrently reducing immunosuppressive factors, including regulatory T cells and myeloid-derived suppressor cells (MDSCs) [6].

Dabrafenib in combination with Trametinib is the most frequently reported treatment associated with DISRs in this category. Of the 36 cases of DISRs analyzed in this review [36,172,173,174,175,176,177,178,179,180,181,182,183,184,185,186,187], 22 occurred in patients receiving this combination therapy. Among these, 21 patients were being treated for metastatic melanoma, while 1 case involved a chiasmatic glioma. The median time to DISR onset was 8.9 months, with a range spanning from 1 to 24 months after the initiation of treatment.

Clinical manifestations were observed in 73% of cases (*N* = 16), with the most frequent being mild cutaneous involvement (13 cases), including erythematous papules and panniculitis lesions. In six cases, patients were asymptomatic. This tendency toward cutaneous involvement mirrors patterns seen with other side effects associated with BRAF and MEK inhibitors, suggesting that skin may represent a primary site of immune-related adverse reactions to these therapies. Notably, there was also a report of one case of uveitis and one case with elevated liver function enzymes [170,171].

Regarding radiological findings, chest CT scans were performed in 9 of the 22 cases. Mediastinal lymphadenopathy was observed in all but one case, which showed isolated parenchymal involvement. It is worth noting that, in the remaining 13 cases, there was no mention of whether radiological investigations were performed. While it is unclear if these imaging studies were carried out, it is likely they were part of routine oncological follow-up protocols. The absence of significant findings could explain why they were not reported.

Diagnoses were confirmed through histopathological analysis, including skin biopsies and EBUS-TBNA, both revealing non-caseating granulomas consistent with DISRs. In the only case without histopathological examination, involving a patient with uveitis, the diagnosis was based on the clinical, laboratory, and instrumental criteria from the 2014 International Workshop on Ocular Sarcoidosis (IWOS). Approaches to patient management differed across cases, with the most common being topical corticosteroids for eight patients and clinical observation without intervention for seven patients. Oral corticosteroids and temporary discontinuation of therapy were used as alternative strategies in selected cases. Only one patient required the permanent discontinuation of therapy. All patients achieved resolution of their DISRs, except for one who experienced partial remission.

While there is no standardized approach to managing these reactions, the frequent spontaneous resolution monitored through follow-up or following treatment with topical corticosteroids suggests that an initial conservative management strategy may be a reasonable choice.

In addition to Dabrafenib and Trametinib, several other BRAF inhibitors have also been associated with DISRs. These include Vemurafenib (eight cases), alone or in combination with Cobimetinib (three cases), Dabrafenib without any MEKi (two cases), and Encorafenib combined with Binimetinib (one case). No significant differences have been noted between these medications and the data concerning Dabrafenib, whether in terms of clinical presentations or management strategies. In addition to the previously reported cases, we identified several case reports in the literature where, although not explicitly labeled as DISRs, histological examinations showed non-necrotizing granulomas [173]. In particular, three cases of interstitial nephritis were identified, all associated with non-necrotizing granulomas found on histological examination. These findings indicate that the occurrence of DISRs related to BRAF inhibitors might be more widespread than what is currently documented in the literature. Increasing awareness of these manifestations could lead to better management strategies and a deeper understanding of their implications (Table 3).

### 3.4. Monoclonal Antibodies

The fourth group was made up of various monoclonal antibodies. Over the past decade, monoclonal antibodies have become widely used, especially in the treatment of chronic inflammatory diseases and lymphomas, because of their precise targeted action. Nonetheless, cases of DISR have been reported in the literature following their use, highlighting how such reactions can occur, even with therapies specifically developed to minimize off-target effects. Specifically, DISR have been observed after the administration of Natalizumab, Tocilizumab, Rituximab, Ustekinumab, Trastuzumab, and Dupilumab [7,188,189,190,191,192,193,194,195,196,197,198,199,200,201,202,203,204,205,206,207,208,209,210,211,212,213,214,215]. Among these, Rituximab accounts for the highest number of reported cases. Despite this association, rituximab is also known to be used as a third-line treatment for sarcoidosis by targeting CD20 proteins on B cells, which play a crucial role in the pathogenesis of this disease. In particular, B cells may contribute by producing autoantibodies, presenting antigens to T cells, and releasing inflammatory cytokines that promote granuloma formation and chronic inflammation [216]. Rituximab-induced sarcoid reactions may result from a disrupted balance in the immune system after B-cell depletion, leading to changes in cytokine regulation and increased T-cell activity. Approximately 6–7 months after Rituximab treatment, as B cells begin to repopulate, the immune environment shifts, with an increase in naive and transitional B cells, higher BAFF (B-cell activating factor) levels, more IL-10-producing B cells, and fewer memory B cells. This shift mimics the immune profile seen in sarcoidosis, which is characterized by similar changes in B-cell subpopulations and cytokine production. As a result, the recovering immune system may inadvertently recreate the inflammatory conditions that contribute to sarcoidosis, potentially triggering DISR [7,201].

In the literature, rituximab has been associated with a total of sixteen patients with sarcoid-like reaction. Twelve of them were treated for non-Hodgkin lymphoma (NHL) as part of a chemotherapy regimen, while four were treated for autoimmune diseases. The average time of the DISR onset over all is fourteen months. However, this value is heavily driven by one outlier exhibiting an onset of 95 months. When dropping the outlier, the average time becomes eight months, which approximately corresponds with the median value too. The most frequent DISR manifestations linked to the administration of monoclonal antibodies are cutaneous and pulmonary. Interestingly, all six patients who developed DISR after being treated with Tocilizumab only exhibited skin involvement. Slightly more than half of the patients experienced general symptoms, primarily exertional dyspnea, joint pain, and skin lesions. The most common imaging finding was hilar lymphadenopathy, although few patients exhibited also a nodular or micronodular pattern.

The treatment of DIRS varied across cases, ranging from the discontinuation of the offending drug to the use of corticosteroids (administered systemically or locally), or clinical follow-up alone. In most cases, the treatment led to a resolution: 28 out of 38 (Table 4).

### 3.5. Miscellaneous Drugs

The fifth and final group encompassed miscellaneous agents not falling into the previous categories. While the number of cases in the literature is small, it is worth noticing that DISR has been also associated with other drug classes, such as interleukin inhibitors (Abatacept) and ALK inhibitors (Lorlatinib) [132,201,217,218,219]. Therefore, it is important to take into consideration the possibility of DISR when administering these drugs (Table 5).

## 4. Diagnosis

Diagnosing DISRs poses significant challenges due to their resemblance to sarcoidosis, other granulomatous diseases, and serious conditions such as malignancies and tumor recurrence or progression, which may necessitate urgent treatment. Misdiagnosis in these cases can lead to inappropriate management and serious consequences. Therefore, a structured approach is essential to differentiate DISRs from these conditions.

Since DISRs are diagnosed by exclusion, the goal is not necessarily to reach a definitive conclusion, but rather to assess the likelihood of the diagnosis by either reinforcing or weakening the initial suspicion. This is performed by identifying supporting evidence while ruling out factors that point to other conditions.

Suspicion of DISRs typically arises from the recognition of characteristic clinical and radiological findings. As observed in the analyzed case reports, these reactions often present with features such as cutaneous lesions, granulomatous infiltration of scars, uveitis, bilateral hilar adenopathy, and increased 18F fluorodeoxyglucose uptake on PET scans. As mentioned earlier, these symptoms may differ based on the specific drug involved, with certain classes of medications being more closely linked to particular clinical presentations.

These manifestations are also common in sarcoidosis, underscoring the importance of a thorough clinical history. For DISRs, this process involves three key steps: establishing a clear temporal association between the start of the drug and the onset of symptoms; reviewing any previous episodes that might suggest undiagnosed sarcoidosis; and ruling out other granulomatous diseases. Examples of such conditions include infectious granulomas, which can result from mycobacterial or fungal infections, and acute granulomatous interstitial lung disease, sometimes associated with methotrexate or Bacillus Calmette–Guérin therapy [220,221,222,223,224]. To facilitate this process, we propose a diagnostic algorithm that aims to clarify the decision-making pathway when confronted with a suspected DISR. The proposed framework is the product of integrating insights derived from previously published algorithms [3] as well as the analysis of the previously discussed case reports. Depending on how well the clinical picture fits, we can classify the likelihood of DISR as unlikely, probable, or highly probable. Figure 5 outlines our proposed diagnostic approach to establishing a DISR.

## 5. Management and Therapy

The effective management of DISRs should consider a variety of clinical and therapeutic factors. As outlined earlier, these conditions often resolve on their own or after stopping the offending drug. In our review, 88% of cases (259 out of 294) showed partial or complete resolution of symptoms and radiological findings spontaneously or after treatment. Only in the remaining 35 cases, no resolution was observed.

These findings suggest that stopping the medication or initiating treatment may not always be necessary. The decision should be based on several key factors, with the most influential being:
The severity of the primary condition;The availability of other therapeutic options;The severity of DISR symptoms.

Primary conditions may differ greatly in terms of severity, progression, and prognosis. For example, in cases where the underlying condition is mild and has a favorable prognosis, it may be simpler for the clinician to consider stopping the drug linked to the DISR. In contrast, for more severe conditions, continuing the current treatment may take precedence.

Another key factor is the availability of alternative treatments; switching to a similar drug in the same class might be ideal in some instances, although the risk of recurrence with a new medication must be carefully evaluated—especially with TNF-alpha inhibitors. When no alternatives are available, clinicians may lean toward maintaining the current regimen. The severity of DISR symptoms, especially in cases involving multiple organs, also plays a significant role. Some DISRs cause minimal or no symptoms, while others show a wider range of clinical manifestations, occasionally affecting multiple organ systems. If the primary drug is effective and DISR symptoms are mild, continuation might be feasible. If the drug is beneficial but the DISR presents significant symptoms, corticosteroids or other treatments used for sarcoidosis could be considered.

Managing DISRs involves several therapeutic options, including careful follow-up, drug discontinuation, topical corticosteroids for localized symptoms, systemic corticosteroids for more widespread involvement, and, when needed, immunomodulatory therapies.

When corticosteroids or other anti-sarcoidosis agents are required for the treatment of a DISR, we suspect that similar doses as those used for sarcoidosis may be effective. However, there is insufficient data to recommend a specific dosing regimen for the disease. Typically, an initial dose of at least 20 mg of daily prednisone equivalent is used. The corticosteroid dose and other immunosuppressants may be adjusted based on the patient’s underlying condition.

Despite these considerations, the current evidence does not yet offer clear recommendations on optimal treatment strategies, such as the recommended duration of clinical observation to allow for spontaneous resolution before initiating additional therapy.

When it comes to follow-up, it is advisable to follow the recommendations from the latest ERS/ATS guidelines on sarcoidosis published in 2020. If future research provides stronger evidence linking specific drug classes to particular clinical manifestations, this may pave the way for developing a more tailored follow-up model for patients with DISRs.

Further studies are crucial to address these gaps in knowledge.

## 6. Conclusions

Drug-induced sarcoid-like reactions (DISRs) represent a rare and poorly understood complication of targeted therapies. They are almost indistinguishable from sarcoidosis in their clinical, radiological, and histopathological presentation, with resolution upon withdrawal of the causative drug remaining the key distinguishing feature.

This close resemblance may, paradoxically, offer an opportunity: drugs known to induce DISRs could provide valuable insights into the immunopathogenesis of sarcoidosis. In this regard, DISRs represent a unique model through which to explore the immune pathways involved in granuloma formation.

Although the current evidence is limited, our case review suggests a possible association between clinical presentation, disease trajectory, and the pharmacological class of the agent involved. These patterns may ultimately contribute to a better understanding of the diverse phenotypes observed in sarcoidosis itself.

Despite this heterogeneity, our analysis confirms that DISRs usually follow a reassuringly benign trajectory: most patients require no specific therapy, and, with careful risk–benefit appraisal, continuation of the targeted agent is feasible. Heightened clinical awareness is therefore essential to avoid misdiagnosis and unnecessary investigations or therapeutic interventions.

Looking ahead, future studies may pave the way for a more standardized yet tailored approach to treatment and follow-up strategies, considering patient-specific factors as well as the characteristics of the causative drug.

## Figures and Tables

**Figure 1 diagnostics-15-01658-f001:**
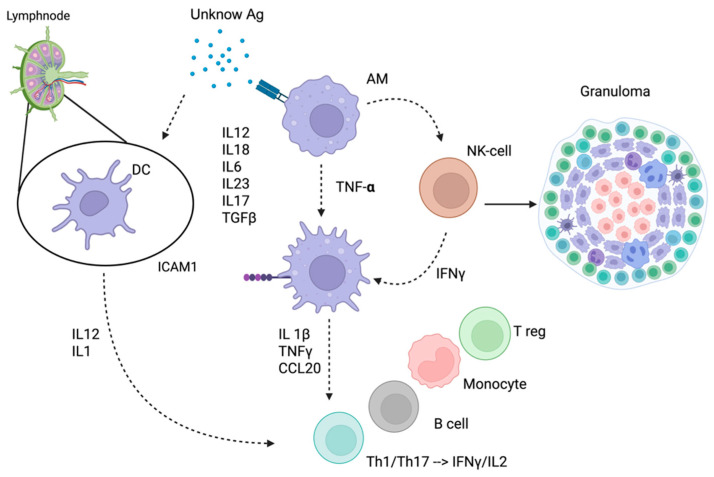
Mechanism of granuloma formation. An unknown antigen (Ag) activates alveolar macrophages (AMs), which begin to produce TNF-α, and dendritic cells (DCs), which migrate to the draining lymph nodes and secrete IL-12 and IL-1. These cytokines promote the differentiation and expansion of Th1 and Th17 lymphocytes, producers of IFN-γ and IL-2, further amplifying the immune response. TNF-α also stimulates natural killer (NK) cells to release IFN-γ, enhancing macrophage activation. Activated macrophages upregulate adhesion molecules (e.g., ICAM-1) and secrete a wide array of pro-inflammatory mediators, including IL-1β, TNF-α, CCL20, IL-12, IL-18, IL-23, IL-17, and TGF-β, contributing to a sustained inflammatory environment. This cytokine-rich environment recruits and organizes specific immune cells, such as Th1 and Th17 cells, B cells, monocytes, and regulatory T cells (Tregs), promoting the formation and maintenance of the granuloma. Created in BioRender. Rubini, S. (2025) https://BioRender.com/c1p8l3p (accessed on 8 May 2025).

**Figure 2 diagnostics-15-01658-f002:**
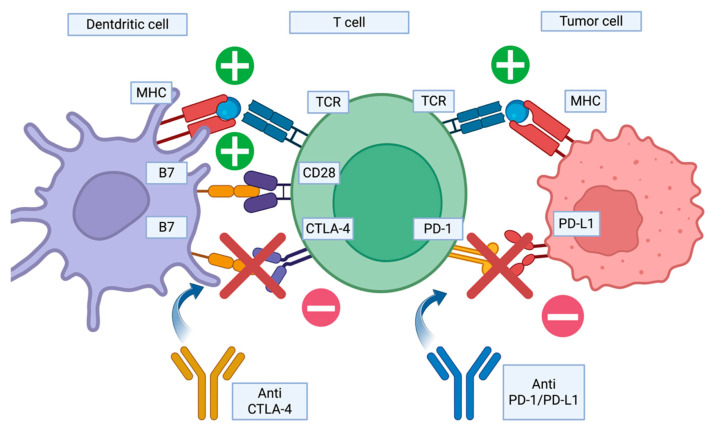
Mechanism of action of immune checkpoint inhibitors (ICIs). ICIs work by blocking inhibitory pathways that suppress immune responses. Ipilimumab targets CTLA-4, which is expressed on activated T cells in lymphoid organs. Nivolumab and Pembrolizumab target PD-1, which is found on T cells in peripheral tissues. Atezolizumab, Durvalumab, and Avelumab target PD-L1, which is expressed on both tumor cells and immune cells. Created in BioRender. Rubini, S. (2025) https://BioRender.com/0s4w3t6 (accessed on 9 May 2025).

**Figure 3 diagnostics-15-01658-f003:**
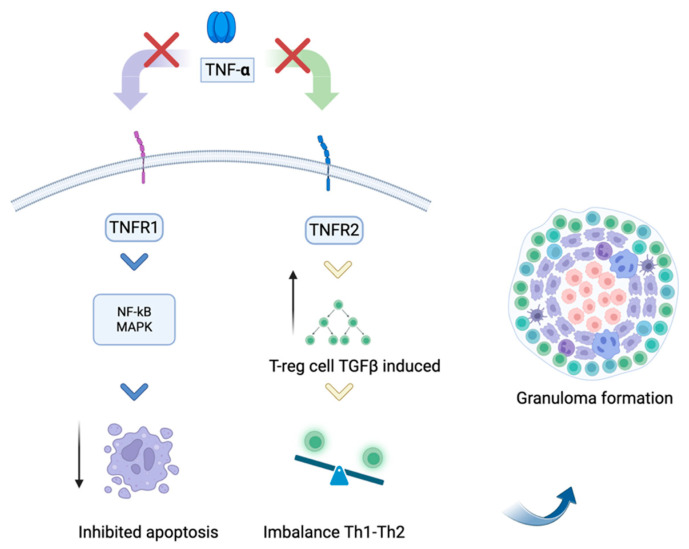
Mechanism of action of anti-tumor necrosis factor (Anti-TNF) agents. Anti-TNF agents selectively bind to TNF-α, inhibiting its interaction with TNFR1 (p55) and thereby preventing the activation of NF-kB, MAPK, and apoptotic pathways. Additionally, they block TNF-α from binding to TNFR2 (p75), leading to a significant increase in Treg cells, which can produce TGF-β, contributing to immune regulation and granuloma formation.

**Figure 4 diagnostics-15-01658-f004:**
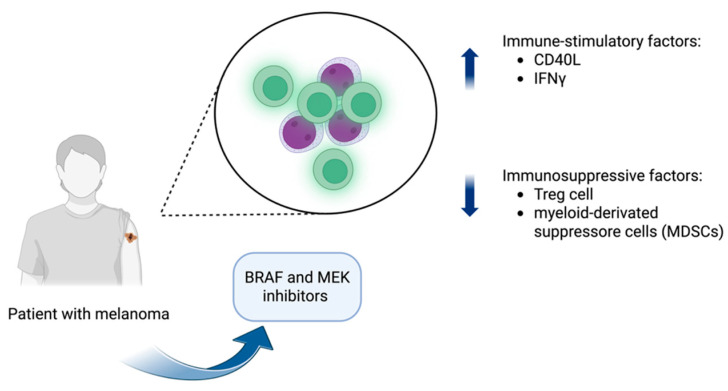
Mechanism of action of BRAF and MEK inhibitors. BRAF inhibitors, often combined with MEK inhibitors, target the mutant BRAF^V600E protein, blocking the MAPK signaling pathway. This leads to altered cytokine profiles, increased expression of immune-stimulatory molecules (e.g., CD40L, IFNγ), and reduced levels of immunosuppressive cells, such as regulatory T cells and myeloid-derived suppressor cells (MDSCs). Created in BioRender. Rubini, S. (2025) https://BioRender.com/6eaaz3x (accessed on 9 May 2025).

**Figure 5 diagnostics-15-01658-f005:**
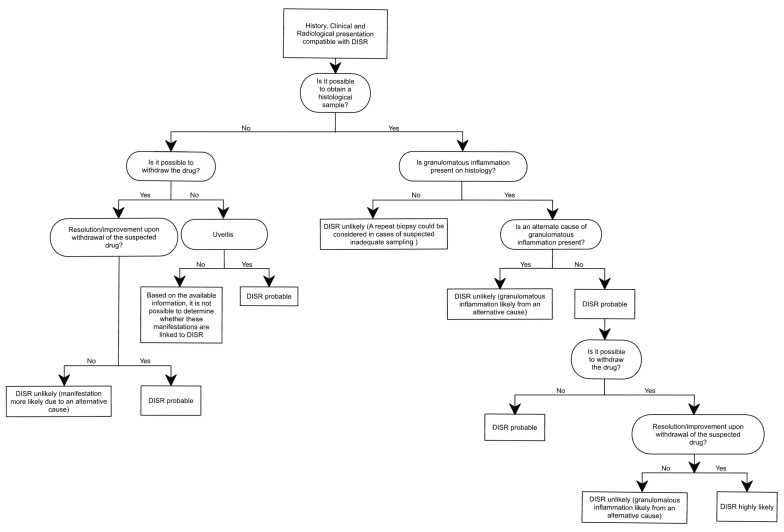
Diagnostic algorithm.

**Table 1 diagnostics-15-01658-t001:** Immune checkpoint inhibitors.

Drug	Mechanism of Action	Treatment Indication	DISR Cases (*n*)	Median Time of Onset (months)	Number of Patients			
					With Signs and Symptoms	Requiring Drug Discontinuation	Requiring Treatment	Showing Resolution
**Pembrolizumab**	Targets and blocks PD-1 ligand	BCA, ccRCC, CRC, cSCC, EAC, NPC, NSCLC, MM, NHL, SARC, UC, ULMS	42	7.32	23 1 NR	17 8 NR	9 8 NR	26 8 NR
**Nivolumab**	Targets and blocks PD-1 ligand	ccRCC, EM, GBM, MM, NSCLC, OM, PC	21	3.9	13 4 NR	10	10	13 4 NR
**Ipilumab**	Inhibits CTLA-4 pathway	MM	20	6.37	15 1 NR	13 2 NR	9 2 NR	18 1 NR
**Nivolumab** **+** **Ipilumab**		CCOC, ccRCC, CRC, HCC, NSCLC, MM, RCC, UC	16	4.15	12 3 NR	5 1 NR	11 1 NR	14 1 NR
**Atezolizumab**	Binds PDL-1 ligand, targeting PD-1 pathway	BCA, HCC, UC	4	3.45	2	3	2	4
**Durvalumab**	Binds PDL-1 ligand, targeting PD-1 pathway	NSCLC	3	NS	2	2	2	3
**Avelumab**	Binds PD-L1 and B7-1 receptors on T cells	OM	1	NS	0	1	0	1
**Sintilimab**	Targets and blocks PD-1 ligand	ESCC	1	NS	1	0	1	1

*Notes:* BCA: breast adenocarcinoma, CCOC: clear cell ovarian cancer, ccRCC: clear cell renal carcinoma, CRC: colon rectal adenocarcinoma, cSCC: cutaneous squamous cell carcinoma, EAC: endometrial adenocarcinoma, EM: esophageal melanoma, ESCC: esophageal squamous cell carcinoma, GBM: glioblastoma, HCC: hepatocellular carcinoma, MM: melanoma, NHL: non-Hodgkin lymphoma, NPC: nasopharyngeal carcinoma, NS: not significant, NSCLC: non-small cell lung cancer, NR: not reported, OM: ocular melanoma, PC: pleiotropic carcinoma, RCC: renal cell carcinoma, SARC: sarcoma, UC: urothelial carcinoma, ULMS: uterine leiomyosarcoma.

**Table 2 diagnostics-15-01658-t002:** TNF-α inhibitors.

Drug	Mechanism of Action	Treatment Indication	DISR Cases (*n*)	Median Time of Onset (months)	Number of Patients			
					With Signs and Symptoms	Requiring Drug Discontinuation	Requiring Treatment	Showing Resolution
**Etanercept**	Binds TNFR2 (p75)	AS, RA, CPP, JIA, JRA, PsA, SS	48 1 (+ ADA) 1 (+ IFX) 1 (+ MTX) 6 (+ PDN)	30	54	36 5 * 2 ** 1 (+ IFX) 1 NR	42 1 NR	55
**Adalimumab**	Blocks TNF-α–TNFR1/2 interaction	RA, AS, BD, CD, CPP, JIA, JRA, PsA, PSO, SAPHO, UC	30 1 (+ ETN)	15.24	28	28 1 ***	27	31
**Infliximab**	Binds to soluble and transmembrane forms of TNF-α	RA, AS, CD, PsA, UC, UP	21	41	17 2 NR	15	13 2 NR	21
**Golimumab**	Binds and inhibits soluble and transmembrane form of TNF-α	AS	1	NS	1	1	1	1
**Certolizumab**	Target the activation of TNF-α	RA	1	NS	1	1	1	1

Notes: ADA: Adalimumab, AS: ankylosing spondylitis, BD: Behçet’s disease, CD: Crohn’s disease, CPP: chronic plaque psoriasis, ETN: Etanercept, JIA: juvenile idiopathic arthritis, JRA: juvenile rheumatoid arthritis, IFX: Infliximab, MTX: methotrexate, NR: not reported, NS: not significant, PDN: prednisone, PsA: psoriatic arthritis, PSO: psoriasis, RA: rheumatoid arthritis, SAPHO: synovitis acne pustulosis hyperostosis and osteitis, SS: Sjogren’s syndrome, UC: ulcerative colitis, UP: ulcerative pancolitis. * Switch to adalimumab; ** switch to tocilizumab; *** switch to infliximab.

**Table 3 diagnostics-15-01658-t003:** BRAF and MEK inhibitors.

Drug	Mechanism of Action	Treatment Indication	DISR Cases (*n*)	Median Time of Onset (months)	Number of Patients			
					With Signs and Symptoms	Requiring Drug Discontinuation	Requiring Treatment	Showing Resolution
**(A) BRAF inhibitors**								
**Vemurafenib**	Competitive inhibitor of BRAF V600E mutation	MM, LCH	8	9	6	3 1 **	4	7
**Dabrafenib**	Selective BRAF inhibitor targeting ATP-binding site	MM	2	NS	2	1 **	1	2
**(B) BRAF inhibitors and MEK inhibitors**								
**Vemurafenib +** **Cobimetinib**	Inhibition of MAPK pathways *	MM	3	NS	3	2 **	2	3
**Dabrafenib +** **Trametinib**	Inhibition of MAPK pathways *	MM, CG	22	8.9	16	3 2 **	12	20
**Encorafenib +** **Binimetinib**	Inhibition of MAPK pathways *	MM	1	NS	1	0	1	1

*Notes:* CG: chiasmatic glioma, MAPK: mitogen-activated protein kinase, MEK: mitogen-activated protein kinase/extracellular signal-regulated kinase, MM: melanoma, NS: not significant, LCH: Langerhans cell histiocytosis. * Interaction as a competitive inhibitor of BRAF V600E mutation + inhibition of MEK1 and MEK2 proteins. ** Drug stopped and then restarted.

**Table 4 diagnostics-15-01658-t004:** Monoclonal antibodies.

Drug	Mechanism of Action	Treatment Indication	DISR Cases (*n*)	Median Time of Onset (months)	Number of Patients			
					With Signs and Symptoms	Requiring Drug Discontinuation	Requiring Treatment	Showing Resolution
**Natalizumab**	Targets the lymphocyte adhesion molecule a4 integrin	CD, MS	3	NS	2	2	2	1
**Rituximab**	Targets CD20 on B lymphocyte cell surface	MCL, CML, DLBCL, NHL, SS, RA, PV, RMPA	16	14	9	0	8	16
**Ustekinumab**	Blocks IL-12 and IL-23	Ps	3	NS	3	3	3	2
**Tocilizumab**	Blocks IL-6	GCA, RA	7	13	2	3	5	6
**Dupilumab**	Blocks IL-4 and IL-13	AD, ERS	3	NS	2	2	3	1
**Trastuzumab**	HER2 receptor inhibitor	Breast cancer	3	NS	1	1	1	1

*Notes:* AD: atopic dermatitis, CD: Crohn’s disease, CML: chronic myelogenous leukemia, DLBCL: diffuse large B-cell lymphoma, ERS: eosinophilic rhinosinusitis, GCS: giant cell arteritis, MCL: mantle cell lymphoma, MS: multiple sclerosis, NHL: non-Hodgkin lymphoma, NS: not significant, Ps: psoriasis, PV: Pemphigus vulgaris, RMPA: refractory microscopic polyangiitis, SS: Sjögren’s syndrome, RA: rheumatoid arthritis.

**Table 5 diagnostics-15-01658-t005:** Miscellaneous drugs.

Drug	Mechanism of Action	Treatment Indication	DISR Cases (*n*)	Median Time of Onset (months)	Number of Patients			
					With Signs and Symptoms	Requiring Drug Discontinuation	Requiring Treatment	Showing Resolution
** (A) Interleukin inhibitor **								
**Abatacept**	Binds to CD80/D86 on antigen-presenting cells, attenuating T-cell activation	RA	1	NS	1	0	1	1
** (B) Tyrosine Kinase inhibitor **								
**Lorlatinib**	Blocks ALK and ROS1 proteins	NSCLC with ALK mutation	2	NS	0	1	0	0

*Notes:* RA: rheumatoid arthritis, NS: not significant, NSCLC: non-small cell lung cancer.

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
