# Peer review of "Drug-Induced Sarcoid-like Reactions Associated to Targeted Therapies and Biologic Agents"

_diagnostics, 2025, doi:10.3390/diagnostics15131658_

Round 1
Reviewer 1 Report
Comments and Suggestions for Authors
This is a well written and structured ms.
The authors have done well to organized their data and structure their study by agents and by molecular mechanisms.
Supplementary metarial is also of high quality.
No revision is required.
Author Response
Comment: This is a well written and structured ms.
The authors have done well to organized their data and structure their study by agents and by molecular mechanisms.
Supplementary metarial is also of high quality.
No revision is required.
Response: We thank the reviewer for his appreciation of our manuscript.
Reviewer 2 Report
Comments and Suggestions for Authors
The paper is devoted to a rather complex problem at the junction of many medical disciplines - drug-induced sarcoid-like reactions (DISR). The authors presented a comprehensive detailed review of this problem. The appearance of such an article is very timely, since today in oncology, rheumatology, pulmonology and other areas new molecules are increasingly used, the side effect of which can be DIRS. The review is written very clearly, distinctly and I have no any concerns.
Author Response
Comment: The paper is devoted to a rather complex problem at the junction of many medical disciplines - drug-induced sarcoid-like reactions (DISR). The authors presented a comprehensive detailed review of this problem. The appearance of such an article is very timely, since today in oncology, rheumatology, pulmonology and other areas new molecules are increasingly used, the side effect of which can be DIRS. The review is written very clearly, distinctly and I have no any concerns.
Response: We thank the reviewer for his appreciation of our manuscript.
Reviewer 3 Report
Comments and Suggestions for Authors
The topic is interesting to a relatively large medical audience given the increasingly number of patients benefiting from advanced treatments but also developing exotic side effects.
Title
The term targeted therapy seems limited – a plural form is probably more adequate; still not all drugs considered in the article are commonly referred as targeted therapies since this expression is closely associated to cancer therapy. TNF inhibitors are not generally referred to as such. Although this may be a pedantic point of view it might be better to rephrase the title.
Abstract – generally fits the article content
Perhaps the results heading could be rephrased as – ‘five main pharmacological groups were associated with DISR in the light of available data ‘ or something similar
Keywords – is there a 6 th keyword missing?
Introduction – tersely presents some physiopathology elements underlying DISR. Perhaps more data on its clinical significance would be welcome
Material and method
The authors described the search methodology and the terms they used; it would be productive to summarize the search results in a PRISMA chart even more so since the exclusion criteria are not clearly stated (which may be acceptable for a narrative review).
The results section is presented as a series of sub chapters – agents, diagnosis, treatment
Perhaps an introduction for each subsection would be useful
- providing some details along the line –ex: we identified five agent groups linked to DISR The first group in the order of importance consisted of immune checkpoint inhibitors…
line 129 – error message possibly from the reference management software – should be removed
Figure 3 from the diagnosis sub section – the box containing ‘DISR probable’ preceding the ‘Is it possible to withdraw the drug’ on the right branch seems redundant. Also ‘highly likely’ should probably be rephrased as ‘highly probable’.
Conclusions seems somewhat generic given the sheer amount of data considered in the article.
The references should be rechecked as there are some duplicates
Please check
ref 8 and 9
57 and 58
125 and 126
241 and 242
Figure 1 and 2 are missing.
Author Response
Comment 1: The topic is interesting to a relatively large medical audience given the increasingly number of patients benefiting from advanced treatments but also developing exotic side effects.
- Title
The term targeted therapy seems limited – a plural form is probably more adequate; still not all drugs considered in the article are commonly referred as targeted therapies since this expression is closely associated to cancer therapy. TNF inhibitors are not generally referred to as such. Although this may be a pedantic point of view it might be better to rephrase the title.
Response 1: We thank the reviewer for this comment: we rephrased the title adding also the term “Biologic Agents” to be more comprehensive.
Comment 2: Abstract – generally fits the article content
Perhaps the results heading could be rephrased as – ‘five main pharmacological groups were associated with DISR in the light of available data ‘ or something similar
Response 2: We rephrased the results heading accordingly.
Comment 3: Keywords – is there a 6 th keyword missing?
Response 3: We thank the reviewer for noticing the mistake. The are no additional keywords. We removed the numbers from keywords list.
Comment 4: Introduction – tersely presents some physiopathology elements underlying DISR. Perhaps more data on its clinical significance would be welcome
Response 4: We acknowledge the comment and we added two sentences in the Introduction to highlight the clinical significance of DISR and the challenges in recognising this condition.
Comment 5: Material and method
The authors described the search methodology and the terms they used; it would be productive to summarize the search results in a PRISMA chart even more so since the exclusion criteria are not clearly stated (which may be acceptable for a narrative review).
Response 5: We appreciate your thoughtful suggestion. As this is a narrative rather than a systematic review, and in line with the reporting standards applicable to this type of article, we did not consider the inclusion of a full PRISMA chart to be warranted. However, we fully acknowledge the value of providing a clear overview of the literature selection process. To that end, we have added what we consider to be the most relevant data points to the brief summary of the search and screening steps, indicating the number of records identified through the initial search strings and ultimately included. In addition, we have made the exclusion criteria used for article selection more explicit.
Comment 6: The results section is presented as a series of sub chapters – agents, diagnosis, treatment
Perhaps an introduction for each subsection would be useful
- providing some details along the line –ex: we identified five agent groups linked to DISR. The first group in the order of importance consisted of immune checkpoint inhibitors…
Response 6: We added a short introduction at the beginning of each subsection.
Comment 7: line 129 – error message possibly from the reference management software – should be removed
Response 7: We apologize not being able to answer to this comment, but we could not find any error at formerly line 129 in our Word file. We wonder if the reviewer refers to a formatting error that might have been generated in the file conversion from Word to PDF. We believe that if the error persists, it could be amended in the proofs.
Comment 8: Figure 3 from the diagnosis sub section – the box containing ‘DISR probable’ preceding the ‘Is it possible to withdraw the drug’ on the right branch seems redundant. Also ‘highly likely’ should probably be rephrased as ‘highly probable’.
Response 8: We have replaced the term "highly likely" with "highly probable". We believe that the box labeled “DISR probable” is intended to emphasize that, even in cases where drug withdrawal is not feasible, the diagnosis of DISR remains probable when all other criteria are met.
Comment 9: Conclusions seems somewhat generic given the sheer amount of data considered in the article.
Response 9: Thank you for your insightful comment. As we did not conduct a formal statistical analysis of the data included in our review, we initially chose to keep the conclusions fairly general. However, in response to your suggestion, we have revised the text to highlight the main findings emerging from our analysis, while clearly emphasizing the need for these observations to be confirmed and further explored through well-designed, targeted studies.
Comment 10: The references should be rechecked as there are some duplicates
Please check
ref 8 and 9
57 and 58
125 and 126
241 and 242
Response 10: We thank the reviewer for this comment. We removed all duplicates and therefore we amended numeration in the manuscript accordingly.
Comment 11: Figure 1 and 2 are missing.
Response 11: We apologize if the reviewer could not see Figure 1 and 2, which appear to be correctly added in our version of the manuscript. We wonder if this is a problem of file compatibility. We will upload also the figures in separate PDF files.